# Modification of Thermo-Chemical Properties of Hot-Pressed ZrB_2_-HfB_2_ Composites by Incorporation of Carbides (SiC, B_4_C, and WC) or Silicides (MoSi_2_ and CrSi_2_) Additives

**DOI:** 10.3390/ma18163761

**Published:** 2025-08-11

**Authors:** Agnieszka Gubernat, Kamil Kornaus, Dariusz Zientara, Łukasz Zych, Paweł Rutkowski, Sebastian Komarek, Annamaria Naughton-Duszova, Yongsheng Liu, Leszek Chlubny, Zbigniew Pędzich

**Affiliations:** 1Faculty of Materials Science and Ceramics, AGH University of Krakow, al. Adama Mickiewicza 30, 30-059 Kraków, Poland; gubernat@agh.edu.pl (A.G.); kornaus@agh.edu.pl (K.K.); zientara@agh.edu.pl (D.Z.); lzych@agh.edu.pl (Ł.Z.); pawelr@agh.edu.pl (P.R.); seko@agh.edu.pl (S.K.); leszek@agh.edu.pl (L.C.); 2Institute of Materials Research, Slovak Academy of Sciences, 47 Watsonova St., 040 01 Košice, Slovakia; aduszova@saske.sk; 3Science and Technology on Thermostructural Composite Materials Laboratory, Northwestern Polytechnical University, Xi’an 710072, China; yongshengliu@nwpu.edu.cn

**Keywords:** HfB_2_, ZrB_2_, composites, additive, carbides, silicides, thermal conductivity, CTE, oxidation resistance

## Abstract

ZrB_2_-HfB_2_ composites allow us to obtain materials characterized by the high chemical resistance characteristic of HfB_2_ while reducing density and improving sinterability due to the presence of ZrB_2_. Since boride composites are difficult-to-sinter materials. One way to achieve high density during sintering is to add phases that activate mass transport processes and, after sintering, remain as composite components that do not degrade and even improve some properties of the borides. The following paper is a comprehensive review of the effects of various and the most commonly used sintering aids, i.e., SiC, B_4_C, WC, MoSi_2_, and CrSi_2_, on the thermo-chemical properties of the ZrB_2_-HfB_2_ composites. High-density composites with a complex phase composition dominated by (Zr,Hf)B_2_ solid solutions were obtained using a hot pressing method. The tests showed differences in the properties of the composites due to the type of sintering additives used. From the point of view of the thermo-chemical properties, the best additive was silicon carbide. The composites containing SiC, when compared to the initial, pure borides, were characterized by high thermal conductivity λ (80–150 W/m·K at 20–1000 °C), a significantly reduced thermal expansion coefficient (CTE ~6.20 × 10^−6^ 1/K at 20–1000 °C), and considerably improved oxidation resistance (up to 1400 °C).

## 1. Introduction

Ultra-high-temperature ceramics (UHTC) are intended for applications as thermal protection in space vehicles and supersonic aircraft [1,2,3,4]. Among ceramic materials included in the UHTC, zirconium and hafnium borides (ZrB_2_ and HfB_2_) are good candidates. They have desirable properties that include a high melting point (>3000 °C), very good chemical resistance, and very good mechanical performance [4,5,6]. These materials have been studied since the 1950s in Russia (previously the USSR) and the USA and are currently experiencing a renaissance of research [7]. Many centers in the USA, China, Italy, and Japan conduct research on the sintering and properties of boride ceramics [2,3,6,7,8,9]. Significant interest in borides of metals such as Ti, Zr, and Hf, as well as Ta, is due to their very good thermal and chemical stability, which creates the possibility of using their borides in extreme environments associated with supersonic flight (>1400 °C in air), space shuttles (>2000 °C in mono-atomic O and N), and rocket engines (3000 °C in reactive chemical vapors). Borides and carbides also show excellent resistance to erosion under the extreme heat flux and gas velocities encountered during supersonic vehicle and space shuttle operations [8].

For use in ultra-high temperature applications, materials with a lower thermal expansion coefficient (CTE) are desirable, as they would minimize the problems of dimensional changes and stress accumulation during heating–cooling cycles. In this perspective, coefficients of linear thermal expansion (α) of the di-borides lie in the range of ~7 to 9 × 10^−6^/K [10,11].

The thermal conductivity of stoichiometric borides ranges from 60 to 120 W/m·K and has a significant effect on heat transfer, leading to temperature equilibration in the material volume and the formation of thermal stresses. Similar to other ceramic materials, the thermal conductivity of single-phase borides depends mainly on the density (porosity) of the polycrystals and grain size [2,3]. In the case of composites of borides of metals of the fourth group of the periodic table of chemical elements, of metal boride-based composites of the fourth group of the periodic table of chemical elements, the heat conduction is significantly affected by the boride sintering activating additives used and phases formed during sintering [4,12].

Oxidation resistance is of extreme importance for materials that are considered for usage under the combined conditions of ultra-high temperatures and ambient atmosphere. Overall, the rate and form of the oxidation primarily depend on the temperature of application, partial pressure of oxygen, gas flow rate, the material density, quality of the material surface, and composition (including the sintering additives/reinforcement contents) [4,13].

Among the di-borides, ZrB_2_ and HfB_2_ exhibit good oxidation resistance at elevated temperatures due to the protective nature of the corresponding oxide layers, with HfB_2_ possibly offering the best oxidation resistance among all transition metal di-boride-based UHTCs [2,3,4]. When di-borides are oxidized at temperatures below 1100 °C, solid metal oxides like MeO_2_ and liquid boron oxide B_2_O_3_ are formed. Exceeding the temperature of 1100 °C leads to evaporation of liquid boron oxide, an increase in porosity, and an increase in the oxidation rate [3,14,15,16,17,18]. The reduction in the oxidation rate becomes possible mainly due to silicon compounds (SiC, disilicides), which are also sintering activators [3,9,16,18,19,20,21]. Owing to these additives, at temperatures higher than 1100 °C, a tight layer of silica or borosilicate glass is formed on the surface, protecting the material from oxygen access to the interior of the material [9,16,18,19,20,22].

The combination of good thermal conductivity, low thermal expansion, and high Young’s modulus should also result in high thermal shock resistance of zirconium–hafnium boride-based composites [4,23,24]. This property is particularly desirable in high-temperature applications.

Among all borides, hafnium boride (HfB_2_) is characterized by one of the best mechanical and chemical properties, while at the same time exhibiting poor sinterability. The composition of the HfB_2_-ZrB_2_ composite provides high strength and good oxidation resistance characteristic of hafnium boride, while reducing the density of the composite and improving sinterability due to the use of zirconium boride. The carbide additives allow the sinterability of the composite to be improved by removing oxide impurities present on the surface of particles of the boride powders through carbothermal reduction. On the other hand, the addition of silicides allows us to obtain composites with a core–shell microstructure with improved mechanical performance [2,3,4,25].

In this paper, the influence of the additive type on thermo-chemical properties of composites basically composed of 40% vol. ZrB_2_, 40% vol. HfB_2_, and 20% vol. of an additive (denoted as MX). For sample densification the hot-pressing method was chosen as the method that could assure the homogeneous microstructure. As the MX additives, carbides such as SiC, B_4_C, and WC, as well as silicides MoSi_2_ and CrSi_2_, were used. Sintering under pressure was carried out at temperatures selected on the basis of the composites’ starting composition, literature analysis, and own experiments [26]. The composites were at first characterized in terms of apparent density, phase composition, and microstructure. Subsequently, measurements of thermo-chemical properties such as thermal expansion, thermal conductivity, and oxidation resistance in the temperature range of 1000–1400 °C were carried out.

## 2. Materials and Methods

The commercially available powders were used to prepare the composites: ZrB_2_, 99.8%, GRADE B, d_50_ = 1.5–3.0 µm (ABCR., Höganäs, Germany); HfB_2_, 99.9%, APS < 1 μm (Nano Research Elements., Dhanora Jattan, India); SiC 99.9%, GRADE UF 25, d_50_ = 0.40–0.65 µm (HC Starck., Goslar, Germany); B_4_C, GRADE HD20, d_50_ = 0.30–0.60 µm (HC Starck., Höganäs, Germany); WC, DS60, FSSS = 0.6–0.7 µm (HC Starck., Goslar, Germany); MoSi_2_, 99.9% (Morton Thiokol Inc., Ogden, UT, USA) and CrSi_2_, 99% (ABCR., Karlsruhe, Germany).

The powders were wet homogenized in ethanol. Alcohol suspensions of the powders were subjected to sonication for 5 min and then mixed for 12 h in a laboratory ball mill with spherical (5 mm) zirconia grinders. The grinders-to-powder-mass ratio was 1:1. After evaporation of the alcohol, the powder mixtures were placed in a graphite die. Samples were hot-pressed (HP) in argon flow using a device made by Thermal-Technology (Minden, NV, USA) equipped with a graphite heating element. Table 1 gives the designations, compositions of the composites, and parameters of the HP sintering process. The hot-pressing pressure was constant through the whole process and was established as 25 MPa. Hot pressing temperatures were chosen according to previous studies and the literature analysis [26].

The sintered samples were disks of 19 mm (¾ inch) diameter and a height of about 5 mm. Apparent density of the composites was determined using the Archimedes method. Their phase composition was determined using the X-ray diffraction method (XRD, Malvern, PANalytical, model Empyrean). Quantitative phase composition of the materials was determined by means of the Rietveld refining using the X’Pert HighScore Plus program (PANalytical, v. 3.0.5.). Databases PDF-2 (released in 2004) and ICSD Database FIZ Karlsruhe (released in 2012) were applied.

The analysis of thermal diffusivity (α) was performed with the laser flash method using the LFA 427 apparatus from Netzsch (Selb, Germany). The measurements were conducted using a 10 mm diameter alumina centering cone and samples with thickness 1.6–2.6 mm, which were placed in a 6 mm thick alumina holder. The measurement was made in 150 mL/min argon flow, 0.6 ms pulse width, and 600 V laser voltage. The laser shots were made at 20, 100, 300, 500, 700, and 900 °C. In order to analyze the collected data, the Proteus v. 4.8.5 Analysis software (Netzsch) was used. For determining the thermal diffusivity (*α*), the Cape–Lehman + ps model was applied. As the reference sample material, pure copper was used to verify data coming from apparatus measurement. The following formula was used to calculate the specific heat capacity (*C_p_*) as a function of temperature (Equation (1)):(1)$$C_{p}=a+b\cdot 10^{-3}T+c\cdot 10^{5}\cdot T^{-2},\;\frac{J}{mol\cdot K}$$
where *a*, *b*, and *c* are the constants shown in Table 2.

The data for the specific heat calculations are collected and presented in Table 2. The initial phase composition of the composites was utilized in the calculations (Table 1).

The value of the thermal conductivity coefficient (λ) was determined according to the following relationship (Equation (2)):(2)$$\lambda =\alpha \cdot \rho \cdot C_{p},\;\frac{W}{m\cdot K}$$
where: 

*α*—thermal diffusivity, *ρ*—apparent density, *C_p_*—heat capacity.

The coefficient of thermal expansion (CTE) was performed by the DIL 402C dilatometer, Netzsch (Germany, Selb). The measurement was performed at the RT-1000 °C temperature range with a 3 K/min heating rate. The experiment was made in 70 mL/min argon flow and at 30 cN pushing rod load. An alumina sample holder was used.

Oxidation resistance of the composites was determined by measuring weight changes of samples heated in an air furnace in the temperature range 1000–1400 °C. Cuboidal samples (10 × 2 × 3 mm) were placed on alumina pads. The heating rate was 5 °C/min, and samples were held at the final temperature of 1000, 1200, or 1400 °C for 2 h. The mass of the samples ranged from 0.5 to 0.8 g, while the surface area was close to 1 cm^2^. The oxidation susceptibility of the materials was presented as the Δm/A dependence (where Δm represents mass changes in g and A is the oxidized sample area in cm^2^) as a function of temperature.

Observations of the composites microstructure, together with a point chemical analysis of the oxidized surface as well as determination of the oxidation profile, were carried out. Metallographic cross-sections of the samples were subjected to SEM observations (Apreo 2, Thermo Scientific., Waltham, MA, USA) combined with the EDS chemical composition analysis.

## 3. Results

This section may be divided by subheadings. It should provide a concise and precise description of the experimental results and their interpretation, as well as the experimental conclusions that can be drawn.

### 3.1. Density Measurements

Table 3 summarizes the results of apparent and relative density measurements of the composites. It also shows the theoretical density of the composites calculated using the rule of mixtures from the volume fractions and theoretical densities of the starting phases.

The hot-pressed composites had high density, exceeding 90% (Table 2). The lowest density of 91.0% was achieved by the reference HP_0 composite without the additives. The density of composites with the carbide additives varied from 98% for the composites with SiC (HP_SC) and WC (HP_WC) additives to 100% for the B_4_C additive (HP_BC). The densities of composites with MoSi_2_ and CrSi_2_ additions were comparable, i.e., 93–94%.

### 3.2. XRD Phase Composition Analysis

The phase composition of the sintered composites is summarized in Table 4. The Table 4 shows solid solutions (Zr,Hf)B_2_ (1) and (Zr,Hf)B_2_ (2) with the same structure but different lattice parameters.

The phase composition of the materials underwent significant changes after sintering (Table 4 and Appendix A). During sintering, a variety of reactions between borides and carbides took place. In the polycrystals: without additives (HP_0), with SiC (HP_SC) and B_4_C (HP_BC) additives, the dominant phase was (Hf,Zr)B_2_ solid solutions, and HfB_2_ and ZrB_2_ borides were also present. A previous paper [26] showed the presence of different solid solutions, i.e., with different guest metal content and different lattice parameters. The phase composition of the WC-containing composite (HP_WC) is much more complex than other composites, containing solid solutions of (Zr,Hf)B_2_ and (Zr,W)C, as well as tungsten monoboride, which was formed by the reaction between ZrB_2_ and WC. The phase composition of composites with CrSi_2_ and MoSi_2_ additions also included (Zr,Hf)B_2_ solid solutions, both borides and silicides, as well as hafnium oxide and silicon oxide (Table 4).

After sintering, in the case of HP_BC, XRD analysis did not detect the presence of boron carbide addition (Table 4). On the basis of SEM microstructural observations (Figure 1 and Appendix A), it can be concluded that the composites obtained are dense and contain boron-rich and carbon-rich precipitates (Appendix A). In order to clarify this observation, an additional phase composition analysis was carried out using a standard. In this way it was possible to determine the presence and the amount of an amorphous phase. It was found that a significant amount of an amorphous phase was present in the HP_BC composite, approximately 20 wt.%, with the rest being a solid solution of (Zr,Hf)B_2_ and hafnium boride HfB_2_ (Appendix A). On this basis, it can be suggested that the boron- and carbon-containing regions visible in the SEM images may be a boron-rich amorphous phase in which carbon is also dissolved.

### 3.3. SEM Microstructure Analysis

Figure 1 (and Appendix A) shows SEM images of microstructures of the composites, which agree with the measured density of composites. The least dense composite was the composite without additives (HP_0), where significant porosity and extensive grain growth can be observed (Figure 1a). The microstructure of other composites was typical for dense materials (Figure 1b–f). The density-dependent gray shades of the various phases present in the material corresponded to the complex phase composition of the composites determined by XRD. In addition, composites with carbide and silicide additions had homogeneous microstructures (Figure 1b–f).

### 3.4. Thermal Conductivity Coefficient (λ)

Figure 2 illustrates the relationship between thermal conductivity and temperature measured for all studied composites. The thermal conductivity coefficient was calculated from the thermal diffusivity (α) and apparent density measurements and specific heat (Cp) calculations (Equations (1) and (2)).

At room temperature, the highest thermal conductivity (λ), close to 100–150 W/m·K, was shown by the composite with SiC and WC additives. In the literature regarding similar material based on ZrB_2_, its conductivity was around 100 W/m·K for samples sintered under pressure at 2000 °C, but a lower amount of SiC, i.e., 20% vol., was used [27]. In that case the grain size was about 5 μm. For the pure hot-pressed ZrB_2_, thermal conductivity was recorded to be 107 W/m·K [28]. The pure HfB_2_ material described in the literature had λ around 117 W/m·K at RT and around 60 W/m·K at 1000 °C, compared to 82 W/m·K for ZrB_2_ [29]. For higher temperatures, the value of thermal conductivity confirms results of the ZrB_2_-HfB_2_ reference solid solution (Figure 2). The paper [30] shows that (Zr,Hf)B_2_ material shows that hafnium introduced to the material structure decreases thermal conductivity from 92 to 82 W/m·K, but it was for only 5 wt.% of hafnium. In the case of the reference sample (HP_0), its thermal conductivity was slightly lower than 80 W/m·K, which can also be caused by the presence of hafnium oxide in the sample, whose thermal conductivity is around 11 W/m·K [30].

For the other composites, the values of the thermal conductivity were distinctly lower and oscillated around 80 W/m·K. In all cases an increase in the temperature caused a decrease in thermal conductivity values. From 500 °C, measured thermal conductivities exhibited no significant changes, which is often the case in multiphase ceramic polycrystals. In the case of the SiC-added composite, the thermal conductivity oscillated around 80 W/m·K, while in the case of the WC-added composite, it was close to 70 W/m·K. The value of thermal conductivity of the other composites, i.e., HP_0, HP_BC, HP_MS, and HP_CS, from 500 °C was around 50 W/m·K. The worst situation of thermal transport was for samples with the introduction of MoSi_2_ and B_4_C and for the reference sample. In the case of reference material, hafnium cations and porosity are responsible for low thermal conductivity.

### 3.5. The Coefficient of Thermal Expansion (CTE)

Figure 3 shows changes in the coefficient of linear expansion of the composites in different temperature ranges. Regardless of the temperature range, the lowest CTE values were shown for composites with SiC (HP_SC), CrSi_2_ (HP_CS), and B_4_C (HP_BC) additives, for which the CTE values ranged from about 5.2 to about 6.5 × 10^−6^ 1/K. The CTE of composites with WC (HP_WC) and MoSi_2_ (HP_MS) ranged from 5.61 to 6.87 × 10^−6^ 1/K. The largest CTE values, ranging from 6.05 to 7.52 × 10^−6^ 1/K, were characteristic for the composite without additives (HP_0).

### 3.6. Oxidation Resistance

Figure 4 presents mass changes related to oxidized ceramics. Mass changes of the samples were recorded at 1000, 1200, and 1400 °C. The mass of samples of most composites did not change much up to 1000 °C. The exception was the composite with the WC addition (HP_WC), in which the mass of the sample noticeably increased. An increase in the oxidation temperature to 1200 °C caused a decrease in the mass of the HP_WC sample, which was even larger at 1400 °C. In the case of the sample with the addition of boron carbide (HP_BC) at 1400 °C, a mass loss was observed. In contrast, a mass gain was seen in HP_0 and HP_CS composites oxidized at 1400 °C, while up to 1200 °C, the mass of these composites changed slightly. A minimal increase in mass was observed for the composite with SiC (HP_SC) already at 1000 °C, but increased oxidation temperature practically did not cause further mass changes. The composite with the MoSi_2_ addition (HP_MS) behaved similarly, but in this case a progressive slight increase in the weight of the composite with an increase in the oxidation temperature was evident.

### 3.7. Oxidation Cross-Sections

Phase composition of the oxidation cross-sections was determined from the XRD phase composition analyses of surfaces of the oxidized composites (Table 5) and spot EDS chemical composition analyses performed during the SEM observations. The cross-sections of the oxidized samples at 1400 °C are shown in Appendix A. Amongst all the composites, the one that stands out was the SiC-added composite, which was covered with a tight and homogeneous SiO_2_ layer of several micrometers (Appendix A). Similar layers can be observed in composites with additions of both thick silicides (Appendix A). Whereby, the layer that covers the ZrB_2_-HfB_2_-MoSi_2_ composite was tight, several micrometers in size, which was about 0.6% of the sample thickness, and contained grains of ZrO_2_ and HfO_2_.

In contrast, the silica layer found on the CrSi_2_-added composite reached 100 µm (about 3% of the sample thickness), was highly degraded, and also contained ZrO_2_ and HfO_2_ grains.

The WC-added composite and the reference composite (HP_0) degraded strongly after the exposure to oxygen. The distinctly porous layer extended deep into the sample to about 500 µm (about 20% of the sample thickness). In both cases, the porous layers were composed mainly of ZrO_2_ and HfO_2_. The composite with the boron carbide additive (HP_BC) was degraded throughout and was composed of ZrO_2_ and HfO_2_ after oxidation (Appendix A).

Detailed maps of the elemental distributions in the oxidation profiles confirming the occurrence of the phases highlighted in Appendix A are shown in Supplementary Appendix A.

## 4. Discussion of the Results

### 4.1. Sintering, Phase Composition, and Microstructure

From the perspective of using boride composites as components of space rockets or supersonic vehicles, the thermal and chemical properties of these materials are important. During the operation of vehicles, stresses arise due to the temperature gradient, an extreme heat flux, and extreme airflow. As a result of friction, vehicle linings are exposed not only to high temperature but also to reactive air components, primarily oxygen. While overcoming the problems associated with producing dense polycrystals through the use of various additives and sintering techniques has been satisfactorily addressed by previous research, producing materials with the desired thermomechanical, thermophysical, thermal, and chemical properties is still a challenge.

In the first part of this work, it was shown that the production of dense composites (Table 2) of ZrB_2_-HfB_2_-MX is possible upon addition of carbide: SiC, B_4_C, and WC, as well as silicides MoSi_2_ and CrSi_2_ [26]. In addition, it was shown that the most homogeneous microstructure is exhibited by all composites sintered by hot pressing (Figure 1). However, it is worth noting that the phase composition of the composites is complex, as shown by the results of XRD analyses summarized in Table 3.

During sintering, various reactions take place between boride, carbide, or silicide additives and oxide impurities. It is also possible that reactive liquid phases from the Si-B-O system are formed during sintering, due to which sintering is activated, with oxide impurities being mostly reduced when carbides are added (Table 3). The presence of liquid phases can be responsible for the formation of microstructures typical of cermets [31,32,33,34,35], which is the case when silicides are added (Figure 1 and Figure 5). The microstructure of composites with MoSi_2_ or CrSi_2_ additives shows grain boundaries and binding phases that can be amorphous [32], as well as hard grain interiors and grain rims that are mainly solid solutions of (Zr,Hf)B_2_, which affects the mechanical and thermal properties of the materials (Figure 5 and Figure 6).

Solid solutions of (Zr,Hf)B_2_ are the main phase present in all composites, regardless of the additive (Table 4). Such a complex phase composition affects the thermal properties of the composites, i.e., thermal conductivity and thermal expansion. From the point of view of the theory describing heat conduction, monocrystals with strong covalent bonds and perfect structure are very good phonon heat conductors; similarly, high thermal conductivity is exhibited by typical electron heat conductors, i.e., metals [36]. In the case of ceramic polycrystals, thermal conductivity depends not only on the nature of the chemical bonds of the dominant phases but also largely on the microstructure of the composites, which includes the grain size of the phases, grain boundaries, and pores. Defects in structure and microstructure, i.e., vacancies, grain boundaries, and pores, as well as the presence of different phases in composites with different densities and structures, are closely related to thermal conductivity and all reduce it [36].

### 4.2. Thermal Conductivity

The boride structure can be described as a sequence of metallic and boron layers of the hexagonal symmetry. The metal layers are densely packed in the main elementary cell, while the boron atoms are in octahedral coordination and are situated in trigonal prisms of the metal lattice. This leads to the formation of a primitive hexagonal crystal lattice, similar to graphite. There are three chemical bonding components in the boride structure: the B-B bond is covalent, there is an ionic component in the Me-B bond, and the Me-Me bond is metallic in nature [3]. Thus, a predominant phonon heat conduction mechanism is expected in borides, but an electron conduction mechanism cannot be excluded. Figure 2 illustrates the temperature dependence curves of the thermal conductivity coefficient. Regardless of the additive used, the heat conduction coefficient of the composites decreases. For both phonon and electron heat conduction mechanisms, an increase in temperature causes interference with phonon–phonon and electron–electron interactions. For single-phase polycrystals, this relationship can be explained by a decrease in the free path of phonons or scattering of electrons on defects as the temperature increases.

The produced composites are characterized by a homogeneous microstructure, but their phase composition is very complex, with solid solutions, oxides, and even amorphous phases present in addition to borides, carbides, and silicides. All these factors affect the value of the thermal conductivity coefficient as a function of temperature. Undoubtedly, the addition of phases that conduct heat well, i.e., SiC or WC, significantly increases the value of the thermal conductivity coefficient compared to the composite without additives, HP_0 (Figure 2). In this case, it can be suggested that the well-conductive phases are continuous and heat conduction follows a percolation mechanism [37]. Significant improvement of thermal conductivity by addition of SiC has been reported in many works [38,39]. As for the other composites, the value of thermal conductivity in the temperature range of 20–200 °C has a significantly lower value of about 80 W/m·K, and as the temperature increases, it does not change as strongly as in the case of HP_SC and HP_WC composites (Figure 2). The composites with the silicide additions show the most complex phase composition; in addition, they may contain amorphous phases, in amounts not determined in the present study. It is most likely that an increase in the number of grain boundaries (planar defects) and the appearance of an amorphous phase will lead to a decrease in thermal conductivity. There are many grain boundaries and interfacial boundaries in the silicide-containing composites. However, it is worth noting that the thermal conductivity of the composites is not significantly worse than that of the HP_0 reference composite. The composite without shows a homogeneous microstructure but is characterized by the largest grain growth. In polycrystals with large grains, there are a small number of grain boundaries, defects on which phonon–phonon interactions are disturbed; from the point of view of heat conduction, grain growth is a favorable phenomenon [40]. The thermal conductivity of pure borides, as well as composites based on ZrB_2_-HfB_2_ with various additives, varies within wide limits from 30 to 150 W/m·K [24,30,37,38,39,40] and depends on the type and amount of additives, on the porosity, and also on the particle size of the powders used. It can be concluded that the produced composites show typical values for boride ceramics in terms of heat conductivity.

High porosity of the composite samples can also be a reason for lower thermal conductivity compared with pure ZrB_2_ material reported in the literature [41]. The influence of porosity on thermal conductivity of materials can be assessed using Klemens’s (Equation (3)) [42] and Maxwell’s (Equation (4)) [43] relations, taking it into account:(3)$$\frac{k}{k_{\alpha }}=1-\frac{4}{3}\varphi$$(4)$$\frac{k}{k_{\alpha }}=1-\frac{3}{2}\varphi$$
where: 

*k*—phonon thermal conductivity of porous material, *k_α_*—phonon thermal conductivity of dense material, *φ*—porosity (inclusion) volume fraction.

For the dense (Zr,Hf)B_2_ material, thermal conductivity should be between 91 and 92.5 W/m·K. The literature data show that the thermal conductivity of the pure (undoped) ZrB_2_ is between 102 and 104 W/m·K, so it is around 12% higher; the calculated results overlap with the conductivity values typically found in the literature. It shows that in the case of the reference sample HP_0, its porosity plays a significant role.

This paper [30] confirms that the introduction of hafnium to the boride structure causes a decrease in crystal lattice volume, which negatively influences the thermal properties of the material. What is important is that the addition of tungsten into the structure causes a higher decrease in the ZrB_2_ lattice parameter and thus much worse heat-transport properties than hafnium addition. The paper confirmed that electron transport dominates thermal conductivity in the case of materials containing transition metal cations. Thus, in our opinion, various transition metals can cause anharmonic lattice vibrations, and in the case of composites with carbide additives, lattice stresses, which decrease the thermal conductivity of the material. Point defects, lattice volume change, and stressed structure can cause electron–phonon scattering as well as phonon–phonon scattering processes. For that reason, the multiphase composition of the investigated composites with various phases, grain size distribution, point defects, and flat defects shows inferior thermal properties compared to the reference sample. In the ZrB_2_-HfB_2_ composite with SiC additive, the increase in thermal conductivity can be caused by large areas rich in SiC phase, which can have 150 W/m·K [28,33,34]. That high thermal conductivity possibly can be caused or explained by a carbon thin film on the grain boundaries, but it has to be examined more in the future. Due to dominant electron–phonon scattering in the boride matrix, phonon–phonon scattering in silicon carbide, and anharmonic vibrations of the lattice, the thermal conductivity decreases rapidly with temperature, reaching around 80 W/m·K at 700 °C. A lower value of λ close to 100 W/m·K was shown by composites with WC and CrSi_2_ additives. In these materials heat conductivity decreases faster with temperature for the CrSi_2_ additive than for the WC additive due to porosity, the presence of oxides, and the quantity of phases with various vibrations of the lattice, so also different electron–phonon scattering and phonon–phonon scattering. The CrSi_2_ phase present in the material shows low thermal conductivity, i.e., around 11 W/m·K, which significantly decreases with temperature [35]. The strong decrease in heat transport in the case of this material can also be caused by the core–shell structure of grains. The change in curve shape (Figure 2) in the case of composites with CrSi_2_ additive is similar to the one recorded for the sample with SiC addition. In the case of material with WC addition, the better thermal properties compared to the reference samples are due to high relative density [36].

In the case of composites, one reason was the fine material microstructure, and the second was the large number of phases present in the material. In the case of the HP_MS sample, a higher amount of hafnium oxide with low thermal conductivity was detected. The MoSi_2_ phase existing in the composite has thermal conductivity around 65 W/m·K at RT [44], which is also responsible for low values of this parameter. The observed core–shell microstructure is composed of ZrB_2_ grains with a shell of (Zr,Hf)B_2_ phase. In accordance with the literature, the addition of hafnium leads to a decrease in lattice volume in (Zr,Hf)B_2_ compared with the ZrB_2_ lattice and thus to a decrease in thermalrmal conductivity. It appears that lower lattice volume leads to stresses and to anharmonic vibrations. In the case of the boron carbide additive, the amorphous phase was present in the composite microstructure, which usually causes a decrease in the material’s thermal conductivity. It is in agreement with the result for B_4_C-containing composites.

### 4.3. Thermal Expansion

The thermal expansion of composites is also affected by a number of factors, which include phase composition, density, porosity, and grain size in the composites [44,45]. In the case of the studied composites, the greatest influence on the value of the linear coefficient of thermal expansion is due to the additives and their effect on the sinterability, microstructure, and phase. Table 6 collects the literature values of the thermal expansion coefficients of the phases forming the composites, but Table 7 summarizes the average values of the thermal expansion coefficient of the composites studied.

From the data presented, it is clear that the addition of carbides with a majority of strong covalent bonds, i.e., low expansion, results in the lowest value of the CTE of the composites at about 6.20 × 10^−6^ 1/K. In contrast, the additives with hexagonal structure (WC, MoSi_2_, and CrSi_2_) characterized by the occurrence of higher CTE anisotropy, compared to the pure borides, slightly reduce or do not change the average value of the thermal expansion coefficient. It is probable that in these cases, the complex phase composition, fine-grained microstructure, and large number of grain and interfacial boundaries reduce the thermal expansion of the composites: HP_WC, HP_MS, and HP_CS. The highest value of the CTE is shown by the reference composite HP_0; in this case, the thermal expansion can be associated with an extensive grain growth.

### 4.4. Discussion of Oxidation Resistance

There are a number of papers describing the oxidation resistance of composites based on ZrB_2_ and HfB_2_ [14,15,16,17,21,50]. Based on the research, it has been established that metal boride particles of the 4th group of the periodic table of chemical elements are covered with passivating oxide layers, i.e., B_2_O_3_ and MeO_2_ [3,4,51]. From the point of view of oxidation, the boron oxide (B_2_O_3_) layer plays an important role. This oxide melts around 450 °C, and as long as it is in a liquid state, it provides protection against the impact of oxygen. Above 1100 °C, the liquid boron oxide changes to the gas phase, according to the reaction (Equation (5)).(5)$$B_{2}O_{3(s)}\overset{450\;{}^{\circ}C}{\to }B_{2}O_{3(l)}\overset{>1100\;{}^{\circ}C}{\to }B_{2}O_{3(g)}$$

The surface of the boride material is revealed and exposed to the destructive influence of oxygen. Although titanium, zirconium, and hafnium dioxides are oxides with high melting points [3,52], they do not effectively protect the interior of the material, and the high partial pressure of B_2_O_3_ contributes to this. Therefore, above 1100 °C, most borides oxidize actively. The described situation occurs in the HP_0 reference composite. In the case of the HP_0 composite oxidized at 1400 °C for 2 h, the depth of the layer damaged by the impact of oxygen and the resulting gaseous B_2_O_3_ is about 500 µm. The surface of the sample is cracked (Figure 7a), and only crystalline zirconium and hafnium oxides are identified on the surface (Table 5, Figure 7a and Appendix A).

The process of active oxidation at temperatures higher than 1100 °C is impossible to stop in the case of the boron carbide composite HP_BC, which is destroyed in a very short time throughout the sample (Figure 7b and Appendix A) [50]. The oxidized surface shows crystallites probably of both oxides, i.e., ZrO_2_ and HfO_2_ (Figure 7b), and areas of very high oxygen concentration, probably amorphous phases. Active oxidation of the composite with B_4_C is favored by the presence of significant amounts of amorphous phase after sintering. The highly reactive amorphous phase reacts readily with oxygen, resulting in strong degradation of the composite (Figure 7b and Appendix A).

The composite with the addition of WC (HP_WC) also shows low oxygen resistance, since very high mass losses from the oxidized surface are observed (Appendix A). A key role in the oxidation of ZrB_2_-HfB_2_-WC composites is played by tungsten carbide [17], which is not resistant to the influence of oxygen. The reaction of WC with oxygen results in the formation of tungsten oxides, primarily WO_3_, which already exhibits high partial pressures at low temperature [17,53,54,55]. When the oxygen partial pressures are high, then the reactions are most likely to occur (Equations (6)–(9)). For these reactions, ΔG is significantly negative already at 200–300 °C [55].(6)$$WC+{2\;O}_{2}\to WO_{2}+CO_{2}↑$$(7)$$WC+{3/2\;O}_{2}\to WO_{2}+CO↑$$(8)$$WC+{2\;O}_{2}\to WO_{3}+CO↑$$(9)$$WC+{5/2\;O}_{2}\to WO_{3}+CO_{2}↑$$

The presence of the WC addition and the formation of volatile tungsten oxides accelerates the oxidation of the composite, increases porosity and thus the possibility of oxygen penetration into the deeper layers of the material, and the formation of gaseous B_2_O_3_, once the temperature exceeds 1100 °C. The ZrB_2_-HfB_2_-WC composite shows a highly degraded surface (Appendix A). The depth of the degraded layer reaches 500 µm (Appendix A). The significant mass loss for this composite is due to the fact that the molar mass of WO_3_ oxide formed by oxidation is three times greater than the molar mass of formed B_2_O_3_ (Appendix A).

The oxidation phenomenon of composites with SiC as well as silicides is different [3,13,18,19,20,56,57,58]. According to the literature data and the presented reaction (Equation (1)), the boron oxide present on the boride particles melts and then, above 1100 °C, transforms into a gaseous state. In silicide (HP_MS and HP_CS) and silicon carbide (HP_SC) composites, silicon plays a key role in the oxidation resistance. In boride composites with SiC addition, it is also possible to form above 1100 °C as a result of the reaction between boron and silicon oxides of borosilicate glass according to reactions (Equations (10) and (11)).(10)$$SiC_{\left(s\right)}+{1.5\;O}_{2(g)}\to SiO_{2(s)}+CO_{\left(g\right)}$$(11)$${B_{2}O_{3}}_{\left(l\right)}+{SiO}_{2(g)}\to B_{2}SiO_{3(s)}+{O_{2}}_{\left(g\right)}$$

Such a composite is protected from oxygen by an outer layer of silica and/or borosilicate glass in which ZrO_2_ and/or HfO_2_ crystals are present. When oxygen access is unlimited, a sufficiently thick SiO2-rich layer is formed, which tightly covers the sample so that the sample interior is protected from degradation. When oxygen access is limited, the resulting SiO_2_ layer is thin, has defects, and oxidation becomes active. On the surface of the HP_SC composite oxidized at 1400 °C (Figure 8a), a tight layer of SiO_2_ and/or B_2_SiO_3_ is visible, with crystals of zirconium oxides or hafnium or zirconium silicate identified by XRD analysis (Table 5). Figure 9 schematically illustrates the oxidation of ZrB_2_-HfB_2_ composites with SiC addition.

The presence of crystalline zirconium and hafnium oxides and zirconium silicate is reasonable. During oxidation, a reaction of zirconium and hafnium borides with oxygen produces stable zirconium and hafnium oxides (Equations (12) and (13)) and liquid boron oxide [19,21]. Further heating leads to the transition of liquid B_2_O_3_ to gaseous B_2_O_3_ (Equation (14)) above 1100 °C. In the case of SiC, silica is formed during oxidation (Equation (15)), and then some of the B_2_O_3_ can dissolve into SiO_2_ (Equation (11)) and form boro-silicate glass [19,57].(12)$${ZrB}_{2(s)}+2.5O_{2(g)}\to {ZrO}_{2(s)}+B_{2}O_{3(l)}$$(13)$${HfB}_{2(s)}+2.5O_{2(g)}\to {HfO}_{2(s)}+B_{2}O_{3(l)}$$(14)$$B_{2}O_{3(l)}\to B_{2}O_{3(g)},\;T>1100{}^{\circ}C$$

EDS chemical analysis further shows how oxygen diffuses inside the HP_SC composite (Figure 10). As can be seen in Figure 10, oxygen diffuses into the interior of the composite along grain boundaries. The grain boundaries can be regarded as defects in the microstructure. From this point of view, oxygen diffusion across grain boundaries in the ZrB_2_-HfB_2_-SiC composite is as likely as possible.

A similar airtight oxide layer on the composite surface is formed when MoSi_2_ is used as an additive (Appendix A) [16,18,19,21]. MoSi_2_ has excellent oxidation resistance at temperatures >1000 °C because of a protective silica surface layer (Equation (16)):(15)$${MoSi}_{2(s)}+{3.5O}_{2(g)}\to 2SiO_{2(s)}+MoO_{3(g)}O_{3(l)}$$

Above 1300 °C, liquid boron oxide can react with MoSi_2_ and oxygen; SiO_2_ and MoB are formed as a result of this reaction (Equation (17)). Mo_5_Si_3_ and SiO_2_ can also be formed (Equation (18)). It is noteworthy that regardless of the formed oxidation products of MoSi_2_, in each case silica is formed, which, forming an airtight layer, protects the interior of the material from further oxidation.(16)$${2MoSi}_{2(s)}+B_{2}O_{3(l)}+{2.5O}_{2(g)}\to 2MoB_{\left(s\right)}+4SiO_{2(s)}$$(17)$${5MoSi}_{2(s)}+{7O}_{2(g)}\to {Mo}_{5}{Si}_{3(s)}+7SiO_{2(s)}$$(18)$${ZrB}_{2(s)}+Mo{Si}_{2(l)}+{2.5O}_{2(g)}\to ZrO_{2(s)}+4SiO_{2(s)}+B_{2}O_{3(g)}+{Mo}_{5}{Si}_{3(s)}$$

In systems with MoSi_2_, oxidation produces stable, high-melting zirconium and hafnium oxides (Equation (19)), whose particles are dispersed in a tight amorphous Si-O-B layer. Figure 8b illustrates the surface of the ZrB_2_-HfB_2_-MoSi_2_ composite, which, according to the data presented, is covered by an SiO_2_ layer (Equations (16)–(19)), in which particles of zirconium, hafnium, and molybdenum oxides can be identified (Equations (16) and (19), Figure 8b).

In addition, for the CrSi_2_ additive, a similar passive oxidation mechanism can be expected as for SiC and MoSi_2_ additives, i.e., the formation of a tight Si-O-B layer (Figure 11). However, as this study shows, the addition of chromium silicide did not increase the oxidation resistance of ZrB_2_-HfB_2_ composites as effectively as SiC and MoSi_2_ additives. Visible in Appendix A, the surface Si-O layer is not as tight as in the composites with SiC addition (Appendix A) and with MoSi_2_ addition (Appendix A). As previous works showed [47,59], various chromium oxides can be formed as a result of oxidation of CrSi_2_ contained in the composites: Cr_2_O_3_, CrO_2_, and CrO_3_ (Equations (20)–(22)).(19)$$2Cr{Si}_{2(s)}+{\frac{11}{2}O}_{2(g)}\to 4SiO_{2(s)}+{Cr}_{2}O_{3(s)}$$(20)$$Cr{Si}_{2(s)}+{3O}_{2(g)}\to 2SiO_{2(s)}+CrO_{2(g)}$$(21)$$Cr{Si}_{2(s)}+{\frac{7}{2}O}_{2(g)}\to 2SiO_{2(s)}+CrO_{3(g)}$$

The formation of CrO_3_ and CrO_2_ is thermodynamically more favorable compared to Cr_2_O_3_ [47,59]. However, the mass gain of the sample during the oxidation (Appendix A) and the results of the XRD analysis of the composite surface after oxidation at 1400 °C (Table 5) indicated the presence of Cr_2_O_3_ only. With limited oxygen supply through a layer of liquid B_2_O_3_ at temperatures not exceeding 1200 °C, the formation of Cr_2_O_3_ is preferred [47,59]. On the other hand, both CrO_3_ and CrO_2_ oxides exist only in the gaseous state, so it is possible that during their volatilization, pores are formed in the SiO_2_ surface layer, which can be observed in Figure 11b and Figure 12. The increase in oxidation temperature from 1200 to 1400 °C causes growth of crystallites of stable oxides of chromium (III), zirconium (IV), and hafnium (IV) (Figure 12).

## 5. Conclusions

The paper presents a comprehensive study of the thermo-chemical properties of composites based on ZrB_2_ and HfB_2_. Dense composites were produced by the hot-pressing technique with the addition of different phases: carbides (SiC, B_4_C, and WC) or silicides (MoSi_2_ and CrSi_2_).

✓These additives allowed the **densification process** to be made easier and were the factor of thermo-chemical property tailoring. The investigated composites showed a complex phase composition dominated by (Zr,Hf)B_2_ solid solutions. The **microstructures** of composites were similar to that observed in cermets containing local core–shell areas. Such a microstructure was mostly present in silicide-added composites.✓The average **coefficient of thermal expansion** of all the composites is lower than the CTE of the reference composite HP_0 and oscillates around 7⸱10^−6^ 1/K. The addition of the carbides in which a strong covalent bond predominates, i.e., SiC and B_4_C, has the greatest effect on reducing the CTE.✓Porosity and excessive grain growth did not affect favorably either **thermal conductivity** or thermal expansion of the reference composite (HP_0). In the case of thermal conductivity λ, the addition of well-conductive phases such as SiC and WC significantly increased the thermal conductivity of the composites. In these cases, a percolation mechanism was anticipated. In the case of the HP_WC composite, a greater increase in conductivity would be expected, but the fine-grained microstructure and the presence of many grain boundaries may adversely affect phonon–phonon interactions.✓**The oxidation resistance** of the composites increased significantly when silicon-containing phases, i.e., SiC, MoSi_2_, and CrSi_2_, were used as the additives. In such a case, when the temperature exceeded 1100 °C, a tight layer of silica or boron–silicate glass formed on the surface, protecting the material from the destructive effects of oxygen. The models of composite oxidation employed in this paper are in agreement with the literature data. Furthermore, it has been shown that in the case of the HP_SC composite, oxygen diffuses deep into the sample along the grain boundaries.✓Among all proposed sintering–activating additives, **silicon carbide** seems to be the best one. With this additive it is possible to obtain composites with high density and values of thermo-chemical parameters desirable from the point of view of high-temperature applications, primarily low thermal expansion, high thermal conductivity, and very good oxidation resistance.

## Figures and Tables

**Figure 1 materials-18-03761-f001:**
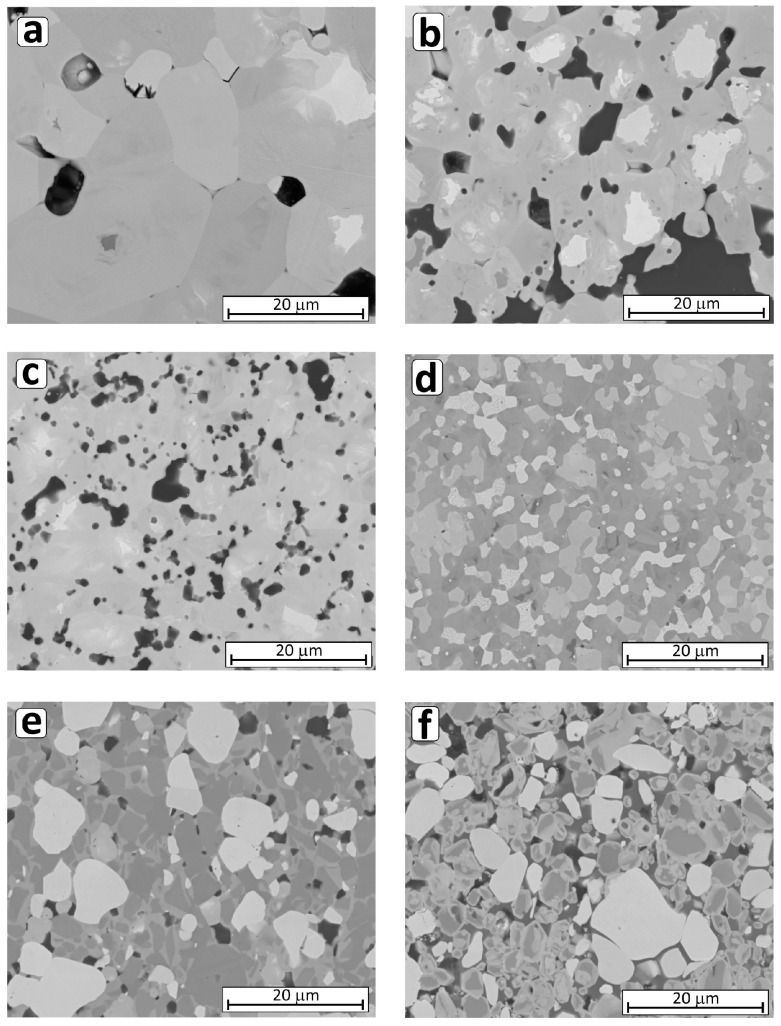
SEM microstructures of ZrB_2_-HfB_2_ composites sintered by HP: (**a**) without additives, (**b**) with SiC, (**c**) with B_4_C, (**d**) with WC, (**e**) with MoSi_2_, and (**f**) with CrSi_2_ additives.

**Figure 2 materials-18-03761-f002:**
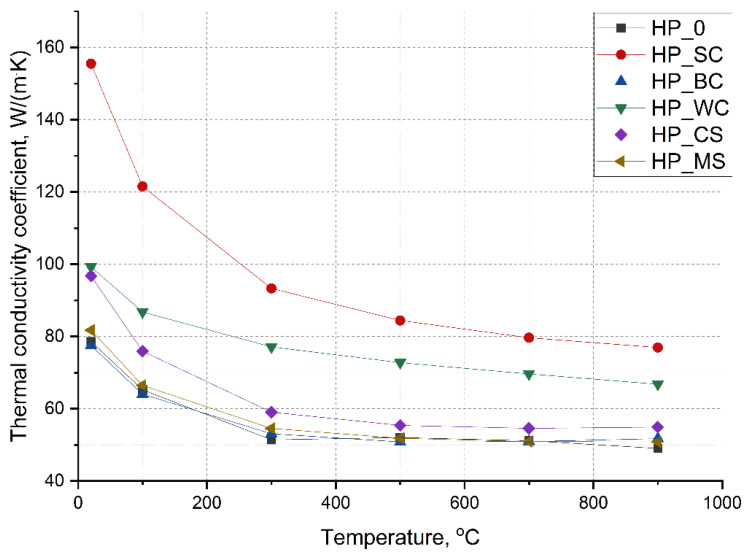
Temperature dependence of calculated thermal conductivity of the composites (the height of the error bars is similar to the size of the markers of the points).

**Figure 3 materials-18-03761-f003:**
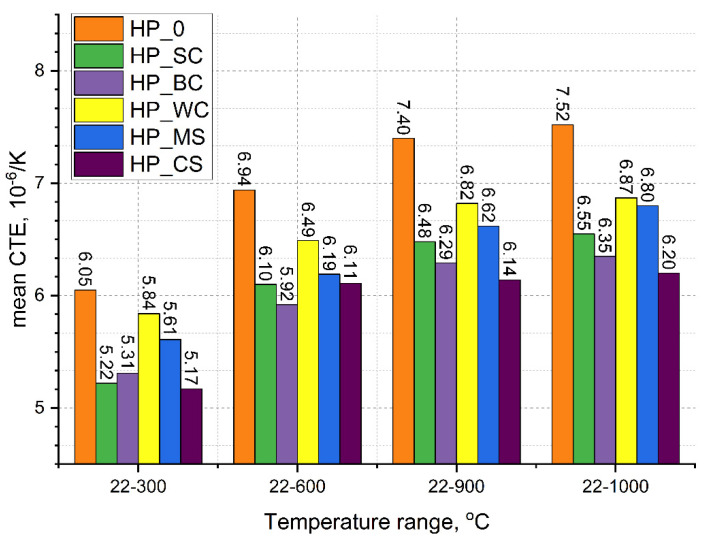
Coefficient of thermal expansion of the composites in different temperature ranges.

**Figure 4 materials-18-03761-f004:**
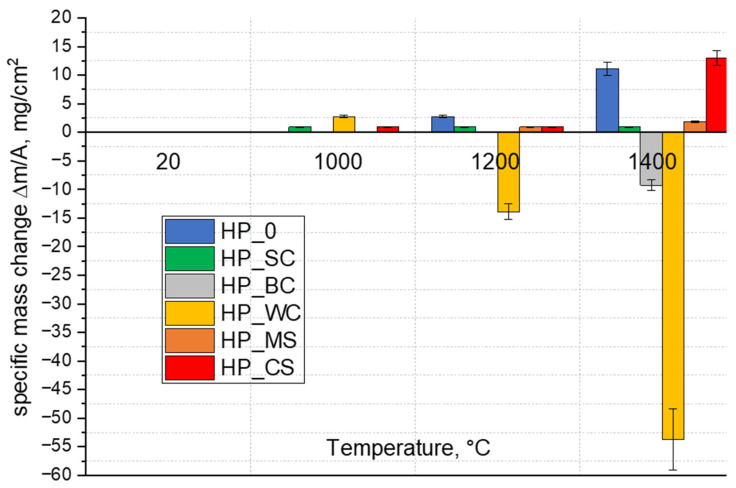
Mass change of the composites heated at 10 °C/min to 1400 °C in air.

**Figure 5 materials-18-03761-f005:**
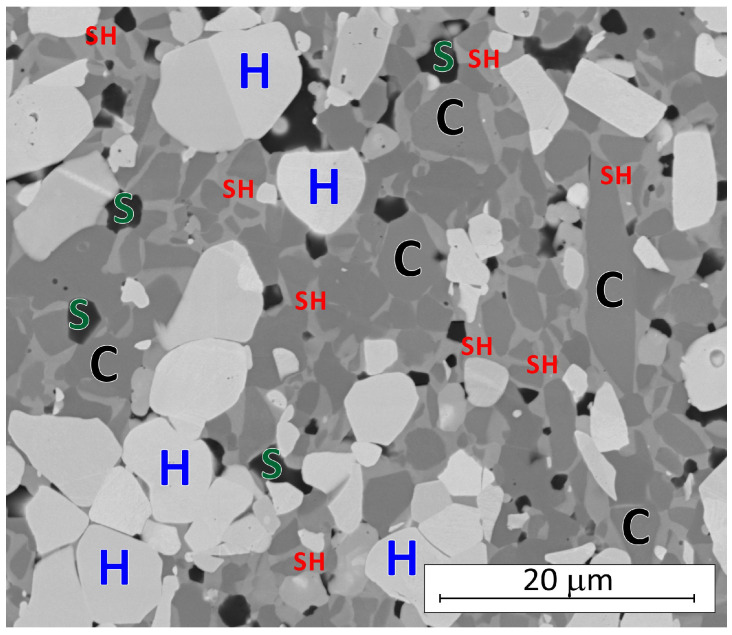
Microstructure of the ZrB_2_-HfB_2_-MoSi_2_ composite (C—core ZrB_2_; SH—shell (Zr,Hf)B_2_; H—HfB_2_; and S—SiC or SiO_2_).

**Figure 6 materials-18-03761-f006:**
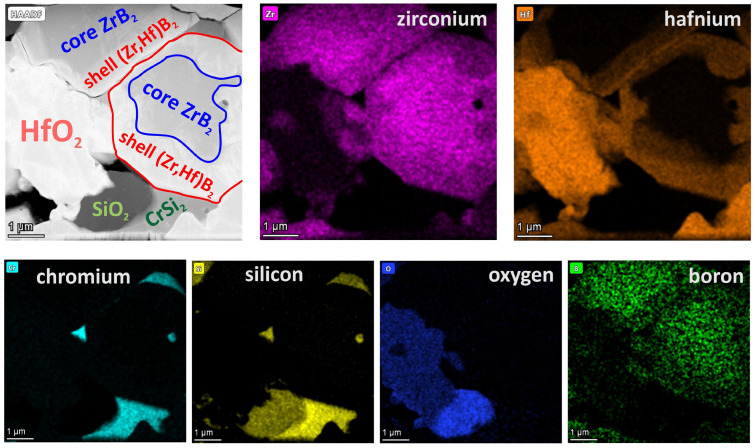
EDS chemical analysis of the HP_CS composite.

**Figure 7 materials-18-03761-f007:**
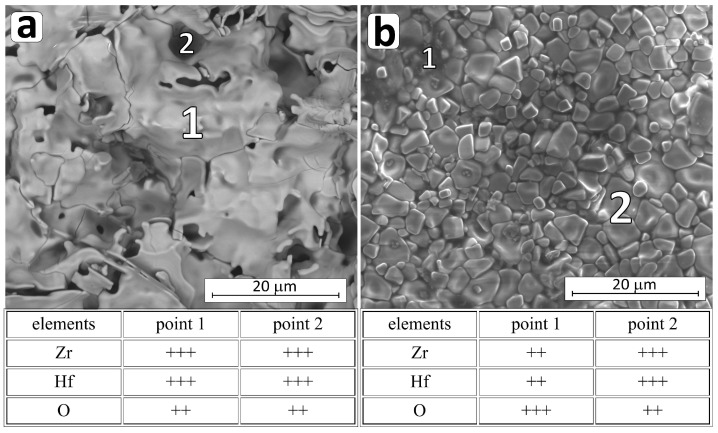
A micrograph of the HP_0 (**a**) and HP_BC (**b**) oxidized composite surface microstructure and chemical composition (EDS) measured in the indicated micro-areas (+++ substantial amount; ++ intermediate amount).

**Figure 8 materials-18-03761-f008:**
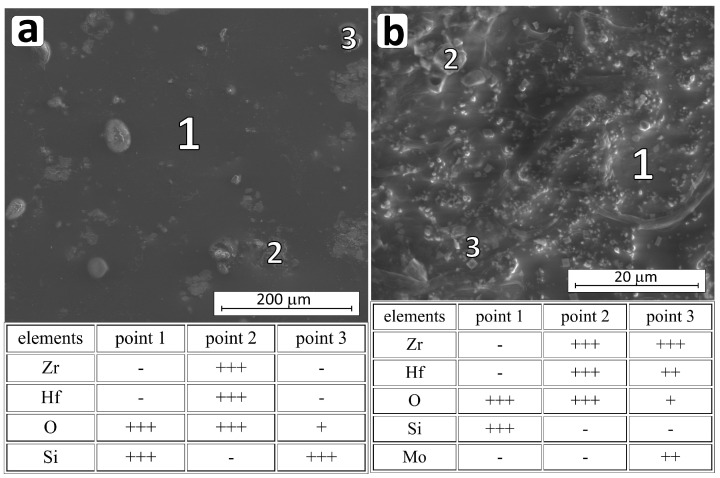
A micrograph of the HP_SC (**a**) and HP_MS (**b**) oxidized composite surface microstructure and chemical composition (EDS) measured in the indicated micro-areas (+++ substantial amount; ++ intermediate amount; + trace amount; - not found).

**Figure 9 materials-18-03761-f009:**
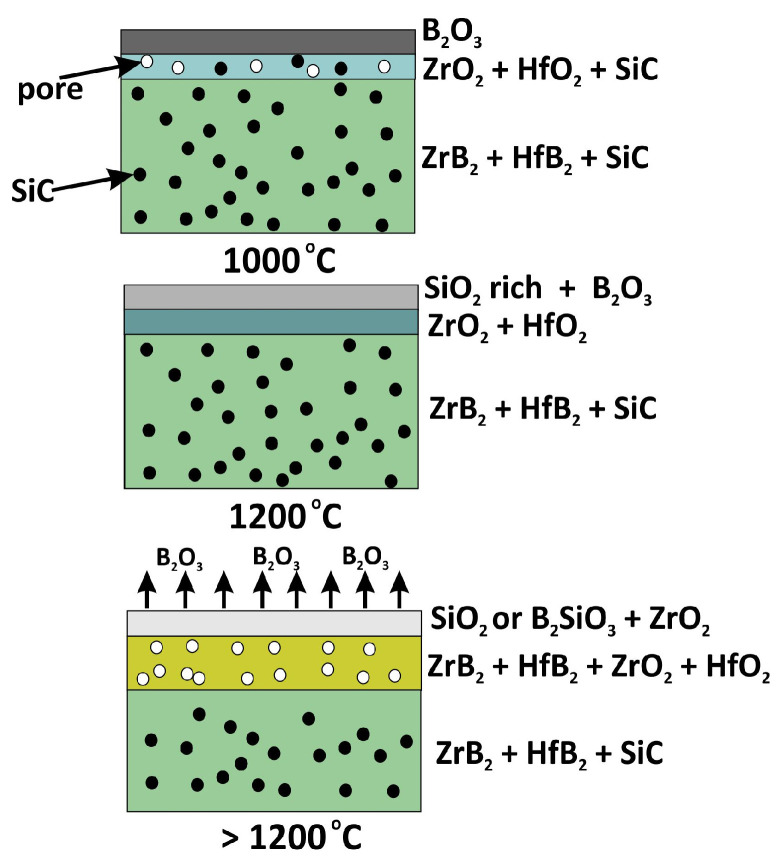
The oxidation scheme of the composite with SiC addition (HP_SC).

**Figure 10 materials-18-03761-f010:**
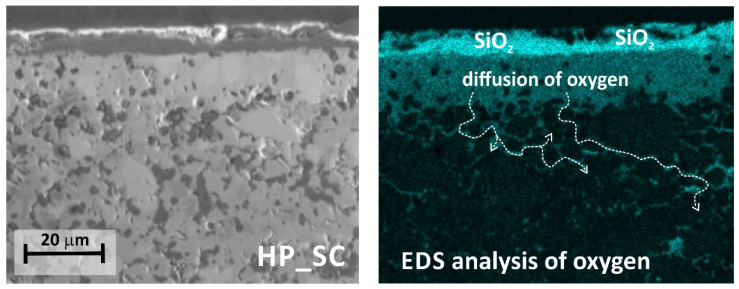
Oxygen diffusion map in HP_SC composite upon oxidation at 1400 °C for 2 h.

**Figure 11 materials-18-03761-f011:**
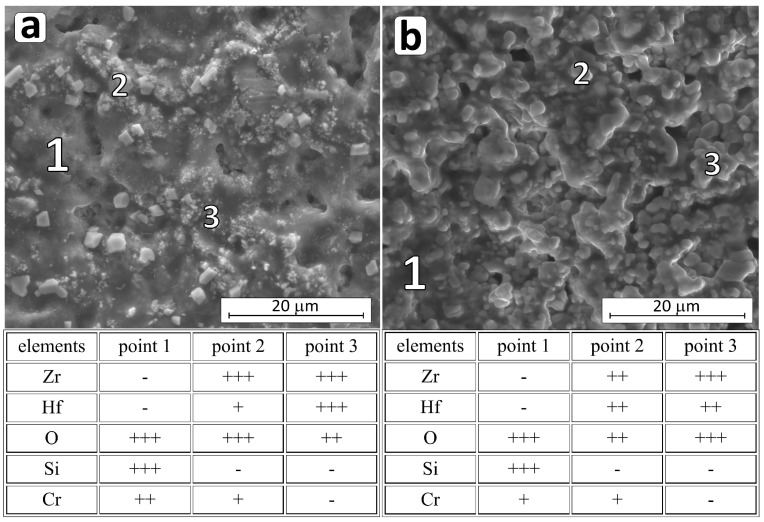
Micrographs of the HP_CS microstructure of the oxidized surface at 1200 °C (**a**) and 1400 °C (**b**) and the chemical composition (EDS) measured in the indicated micro-areas (+++ substantial amount; ++ intermediate amount; + trace amount; - not found).

**Figure 12 materials-18-03761-f012:**
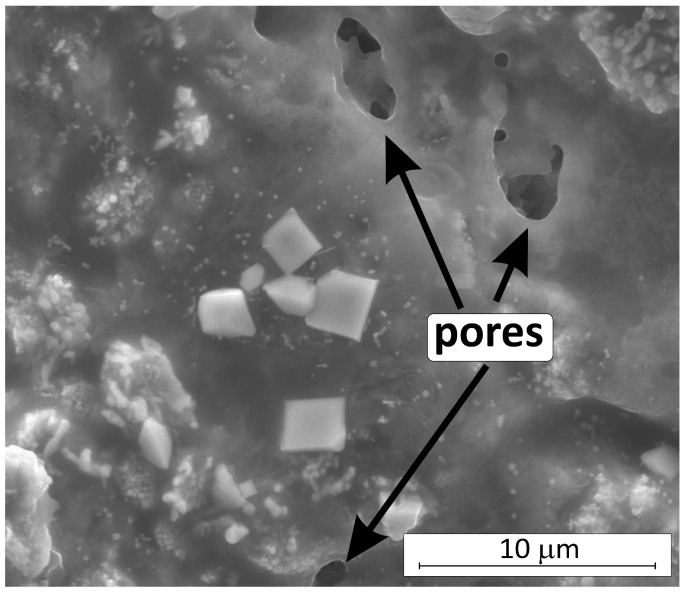
The visible pores in the Si-O-B-Cr layer on the surface of the oxidized HP_CS composite.

**Table 1 materials-18-03761-t001:** Composition, hot-pressing conditions, and designation of composites.

Designation of the Composite	Composition, % vol.			Sintering Temperature, °C	Dwelling Time, h	Heating Rate, °C/min
	ZrB_2_	HfB_2_	Addition			
HP_0	50	50	-	2100	1	10
HP_SC	40	40	20 (SiC)	2000		10
HP_BC	40	40	20 (B_4_C)	2000		10
HP_WC	40	40	20 (WC)	2000		10
HP_MS	40	40	20 (MoSi_2_)	1750		10
HP_CS	40	40	20 (CrSi_2_)	1550		10

**Table 2 materials-18-03761-t002:** Constants for the calculation of heat capacity *C_p_* of selected borides, carbides, and silicides [10].

Compound	*a*	*b*	*c*	Temperature Interval, K
ZrB_2_	62.300	23.012	17.489	298–3310
HfB_2_	73.764	7.824	23.012	298–3520
SiC	41.714	7.615	15.230	298–3100
B_4_C	96.190	22.594	44.852	298–1200
WC	51.33	8.619	11.213	298–3060
MoSi_2_	67.488	15.523	7.406	298–1200
CrSi_2_	59.831	44.05	174.724	298–873

**Table 3 materials-18-03761-t003:** Theoretical and experimental density of the sintered ceramics as measured by Archimedes method.

Composite	Theoretical Density *, g/cm^3^	Apparent Density, g/cm^3^	Relative Density, %
HP_0	8.29	7.55	91.0
HP_SC	7.25	7.16	98.4
HP_BC	7.24	7.24	100.0
HP_WC	9.79	9.64	98.5
HP_MS	7.88	7.37	93.5
HP_CS	7.62	7.18	94.4

* Calculated value from the volume fractions and theoretical densities of the starting phases.

**Table 4 materials-18-03761-t004:** Results of phase composition analyses of the composites.

Composite	Quantitative Phase Composition, wt%
HP_0	94.1% (Zr,Hf)B_2_, 3.3% ZrB_2_, 2.6% HfO_2_
HP_SC	71.8% (Zr,Hf)B_2_, 9.7% HfB_2_, 13.4% SiC, 5.1% ZrB_2_
HP_BC	98.9% (Zr,Hf)B_2_, 1.1% HfB_2_
HP_WC	18.5% (Zr,Hf)B_2_ (1), 16.5% (Zr,Hf)B_2_ (2), 11.9% ZrB_2_, 4.2% HfB_2_, 24.8% (Zr,W)C, 21.5% WB
HP_MS	23.1% (Zr,Hf)B_2_, 28.8% HfB_2_, 25.2% ZrB_2_, 17.9% MoSi_2_, 5.2%HfO_2_
HP_CS	38.4% (Zr,Hf)B_2_ (1), 27.7% (Zr,Hf)B_2_ (2), 18.6% ZrB_2_, 7.9% HfB_2_, 0.3% SiO_2_,6.3% CrSi_2_, 0.9% HfO_2_

**Table 5 materials-18-03761-t005:** Crystalline phases identified on the surface of the composites oxidized at 1400 °C.

Composite	Mass Change, mg/cm^2^	Thickness of the Oxidated Layer, μm (% of Thickness)	Phases by XRD
HP_0	+11	500 (20.0%)	m-ZrO_2_, HfO_2_
HP_SC	+1	10 (0.3%)	ZrSiO_4_, m-ZrO_2_
HP_BC	−10	full volume oxidation (100%)	m-ZrO_2_
HP_WC	−54	500 (20.0%)	m-ZrO_2_, WO_x_, HfB traces
HP_MS	+2	20 (0.6%)	ZrSiO_4_, m-ZrO_2_, cristobalite, MoB
HP_CS	+13	100 (3.0%)	ZrSiO_4_, Cr_2_O_3_, cristobalite, traces of t-ZrO_2_

**Table 6 materials-18-03761-t006:** The coefficient of thermal expansion of the constituent phases of the composites.

Compound	CTE, ×10^−6^ 1/K	Reference
ZrB_2_	5.9–6.8 (300–2300 K)	[46]
HfB_2_	6.3–6.8 (300–2300 K)	
SiC	5.1–5.9 (300–2500 K)	[11]
B_4_C	4.8–6.5 (300–2300 K)	[11]
WC	axis a 5.2, axis c 7.3 (300–2100 K)	[11]
MoSi_2_	7–10 (300–1300 K)	[47]
CrSi_2_	7.4–9.2 (300–900 K)	[48,49]

a, c—crystallographic axes of WC unit cell.

**Table 7 materials-18-03761-t007:** The average value of the coefficient of thermal expansion of the tested composites.

Composite	Mean CTE, ×10^−6^ 1/K 300–1200 K
HP_0	7.52
HP_SC	6.55
HP_BC	6.35
HP_WC	6.87
HP_MS	6.80
HP_CS	6.55

## Data Availability

The data presented in this study are available upon request from the corresponding author.

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
