# Peer review of "Modification of Thermo-Chemical Properties of Hot-Pressed ZrB2-HfB2 Composites by Incorporation of Carbides (SiC, B4C, and WC) or Silicides (MoSi2 and CrSi2) Additives"

_materials, 2025, doi:10.3390/ma18163761_

Round 1

Reviewer 1 Report

Comments and Suggestions for Authors

1) Some of the relative density values in table 3 seem very unrealistic. 98% or 100% density are really uncommon. There might be inconsistencies with how you measured the density with the use of the Archimedes method. Please recalculate.

2) Please add the X-ray diffractograms in section 3.2 for the different studied composites.

3) Please add another figure with SEM images and elemental maps for the different studied composites as sintered. This would help expand the discussion.

4) I would highly suggest to add the relevant SEM images of the cross sections to section 3.7. This would help to improve this work.

5) Discussion of the results is very long. You might want to break it down to different sections. Another alternative would be to include the discussion in chapter 3 along with the results.

6) It would be beneficial to use bullet points for the conclusions 

My closing comment is that this is a really good work, with interesting results. Some minor changes and organising results and discussion would help to improve this manuscript.

Author Response

Thank you very much for your review. We appreciate all your comments. Below are our explanations and comments on your remarks.

Comments 1: Some of the relative density values in table 3 seem very unrealistic. 98% or 100% density are really uncommon. There might be inconsistencies with how you measured the density with the use of the Archimedes method. Please recalculate.

Response 1: Results of the calculations of relative density of the composites are just estimates. This is because of the difficulty in accurately calculating the theoretical density. We based the calculations on an initial composition of the composites. During their sintering, a series of solid state reactions occurred, resulting in new phases, and formation of an amorphous phase. Therefore, the calculation of the theoretical density based on phase composition analysis of the sintered materials is also subject to significant error. We added an appropriate explanation in the text.

Comments 2: Please add the X-ray diffractograms in section 3.2 for the different studied composites.

Response 2: Thank you for your comment. The article is very long, so we suggest adding the diffractograms to the appendix.

Comments 3: Please add another figure with SEM images and elemental maps for the different studied composites as sintered. This would help expand the discussion.

Response 3: Unfortunately, we do not have elemental distribution maps for all samples. Most of the samples were used to test various characteristics. The final stage of the samples characterization was the test of oxidation resistance, and as we know from the work, after this test the samples were not suitable for the repetition of e.g., SEM observations. In the supplementary material we included the descriptions of the microstructures based on the point-by-point EDS chemical composition analysis. We hope that it can provide a substitute for the elemental distribution maps.

Comments 4: I would highly suggest to add the relevant SEM images of the cross sections to section 3.7. This would help to improve this work.

Response 4: In the original version of the work, oxidation cross-sections were added. Currently, the cross-sections, together with the elemental distribution maps, can be found in the supplementary material.

Comments 5: Discussion of the results is very long. You might want to break it down to different sections. Another alternative would be to include the discussion in chapter 3 along with the results.

Response 5: The discussion of the results has been divided into sections.

Comments 6: It would be beneficial to use bullet points for the conclusions 

Response 6: The conclusions have been summarised.

Reviewer 2 Report

Comments and Suggestions for Authors

Comments and suggestions for authors are attached in the word file.

Author Response

Thank you very much for your review. We appreciate all your comments. Below are our explanations and comments on your remarks.

Comments 1: From 500°C, measured thermal conductivities exhibited not significant changes what is a typical phenomenon. Any literature regarding this?

Response 1: Thank you for your comment. We agree that this statement is imprecise. According to the theory, the higher the temperature, the more phonons with short wavelengths participate in heat conduction. The mean free path of  phonons is reduced because there is a significant contribution of phonon scattering at defects, impurities and grain boundaries, regardless of the type of polycrystalline sample. Therefore, changes in the values of the thermal conductivity coefficient are not as distinct as at temperatures below 500°C. We have revised this sentence.

Comments 2: Can authors explain, why is sample HP_BC is disintegrated ?

Response 2: Thank you very much for your comment. ‘Disintegrated’ is too strong a term, because actually the HP_BC sample did not disintegrate, but underwent chemical corrosion (oxidation) throughout its entire volume. Above 1100°C, boron oxide - B2O3, which passivates boron and boron carbide particles, transforms into a gaseous phase. Fresh surfaces of the sample are exposed, and as a result of oxidation, boron carbide changes into a gaseous phase (according to the reaction below) and hafnium and zirconium oxides, which are stable under these conditions, are formed.

B4C+4O2=2B2O3(g)+CO2(g)

The active oxidation of the HP_BC sample is also facilitated by the presence of significant amounts of a reactive amorphous phase.

Comments 3: Sentence: As for the other composites, the value of thermal conductivity in the temperature range of 20-200°C has a significantly lower value of about 80 W/m·K, and as the temperature increases, it does not change as strongly as in the case of HP_SC and HP_WC composites (Fig. 3). Comment: This is not fig. 3 it is fig. 2.

Response 3: This error has been corrected.

Reviewer 3 Report

Comments and Suggestions for Authors

1.- 1.- The first and main problem is that the entire first part has already been reported by the authors, including the same tables and figures (Table 1 and 2, with some minor modifications, as well as Figure 1) at https://doi.org/10.1016/j.jeurceramsoc.2024.116685, so the way they are presenting the results cannot be acceptable. It is understood that the thermal properties of the sintered materials have now been studied, but the results cannot be presented in the present form. Therefore they should be rewritten in such a way that there is no doubt which part is being reported.

2.- How was the thickness of table 5 measured?
3.- The phrase “It is most likely that an increase in the number of grain boundaries (planar defects) and the appearance of an amorphous phase will lead to a decrease in thermal conductivity” is problematic because it cannot be demonstrated with the reported results. 

4.- The entire section on pages 12-14 is highly speculative: since the authors do not present complete characterizations of the systems analyzed, they focus on providing explanations of references that do not necessarily correspond to their analyzed systems.
5.- Enter the delta G values for the reactions.

Author Response

1.- 1.- The first and main problem is that the entire first part has already been reported by the authors, including the same tables and figures (Table 1 and 2, with some minor modifications, as well as Figure 1) at https://doi.org/10.1016/j.jeurceramsoc.2024.116685, so the way they are presenting the results cannot be acceptable. It is understood that the thermal properties of the sintered materials have now been studied, but the results cannot be presented in the present form. Therefore they should be rewritten in such a way that there is no doubt which part is being reported.

Dear Reviewer, we believe this accusation is unfounded and unethical. The article in JESC concerns the manufacturing of UHTC composites using various sintering techniques, i.e., HP, SPS, and HPHT. The article describes sintering and selects the best technique. The article submitted for review in Materials describes studies on the thermal and chemical properties of composites produced using the HP technique, which we consider the best. The reviewer claims that Tables 1 and 2 are the same. While Table 1 in both papers is the same because it describes the same composites, Table 2 in JECS concerns composites sintered using the SPS technique, while Table 2 in Materials describes specific heat calculations, etc. Figure 1 in Materials shows the microstructures of composites sintered using the HP technique, while Figure 1 in JESC shows the diffraction patterns of HP_0 and HPHT_0 composites, therefore they are not the same. In our opinion, making such inappropriate accusations requires thorough verification.

Table 1 has been amended.

2.- How was the thickness of table 5 measured?

We measured the thickness of the oxidized layer using an SEM microscope.

3.- The phrase “It is most likely that an increase in the number of grain boundaries (planar defects) and the appearance of an amorphous phase will lead to a decrease in thermal conductivity” is problematic because it cannot be demonstrated with the reported results. 

In this article, we present large-scale microstructure images (Fig. 1), which demonstrate particle size, interphase boundaries, and intergranular boundaries. In the case of the HP_WC composite, the number of intergranular and interphase boundaries is significant due to its fine-grained microstructure. In the case of the HP_BC composite, we demonstrate the presence of an amorphous phase in the supplementary material.

4.- The entire section on pages 12-14 is highly speculative: since the authors do not present complete characterizations of the systems analyzed, they focus on providing explanations of references that do not necessarily correspond to their analyzed systems.

We do not understand this objection either, the description of heat conduction provided refers to the composites we tested.

5.- Enter the delta G values for the reactions.

We are unable to calculate delta G for all reactions. We conducted a thorough literature review. There are papers that provide delta G calculations. The delta G values presented in these papers differ very much for the same reactions. In our discussion of oxidation, we presented the most probable reactions.

Reviewer 4 Report

Comments and Suggestions for Authors

The article "Modification of thermo-chemical properties of hot-pressed ZrB2-HfB2 composites by incorporation of carbides (SiC, B4C, WC) or silicides (MoSi2, CrSi2) additive" is submitted for publication in the journal "Materials". The work is presented in clear language, well illustrated, the essence is fully disclosed. The introduction is presented in sufficient detail, it sets out the history of the issue and the essence of the research well. The goal is clearly stated. The Materials and methods section describes them in detail. The methods and equipment are appropriately selected. The conclusions correspond to the work done. Thus, I recommend the article to be published after minor revisions:

  1. The size ruler in Figures 1, 6, 7, 8 is hard to read, please correct.
  2. I ask you to make figures 2,3,4 in a single style.

Author Response

Thank you very much for your review. We appreciate all your comments. Below are our explanations and comments on your remarks.

Comments 1: The size ruler in Figures 1, 6, 7, 8 is hard to read, please correct.

Respone 1: The scale bars in Figures 1, 6, 7, 8, and others have been corrected.

Comments 2: I ask you to make figures 2,3,4 in a single style.

Response 2: The graphs in Figures 2,3,4 have been done in one style.

Round 2

Reviewer 3 Report

Comments and Suggestions for Authors

The authors corrected the manuscript and it can be published.